SOFTWARE

# A methodology to reduce the localization error in multi-loci microscopy provides new insights into enhancer biology

**Christopher H. Bohrer, Daniel R. Larson**⬤*

Laboratory of Receptor Biology and Gene Expression, National Institutes of Health, Bethesda, Maryland, United States of America

* Dan.larson@nih.gov

## Abstract

Numerous functions hinge on the spatial arrangement of different genomic loci. Hence, microscopy techniques, such as chromatin tracing, have been developed to localize multiple loci in fixed cells. Depending on the throughput and specifics of the experiment, localization errors can still obscure the true spatial locations. We have developed a post-processing methodology to address this challenge without the need for additional experimentation: Loci Enabled Advanced Resolution (LEAR). By leveraging the fact that localization errors increase the variability of the displacements between loci, and given an approximation of the localization error, we can approximate the ground truth spatial variation for each pair of loci to guide an iterative error correction process. After validating our approach with simulation and experiment, we then applied our approach to existing chromatin tracing data that probed the relation between chromatin organization and *Sox2* regulation, where previous work found no clear correlation between enhancer-promoter proximity and transcription bursts in individual cells. We discovered a correlation previously obscured by localization error, clearly demonstrating the need for the methodology. We then investigated the influence of loop-extrusion on higher order multi-way contact frequencies, which dramatically increased with the application of the LEAR method, finding that certain multi-way contacts were only present with loop-extrusion.

## Introduction

Multiplexing, the use of multiple colors, and the capability to re-image the same biological sample enables the imaging of numerous distinct targets within individual cells [3,37,73]. A key development within this vein is chromatin tracing [6–8,22,30,33,35, 48–50,52–54,63]. Chromatin tracing is an extension of traditional DNA FISH [37] that involves hybridizing probes, imaging, washing, and then repeating the process with different DNA targets [11,69], allowing the identification of multiple loci without solely relying on color differentiation. Chromatin tracing can now visualize thousands of

**Data availability statement:** All data and code for this work are available at: https://github.com/CHB-Bohrer/.

**Funding:** This research and salaries of DRL and CHB were supported by the Intramural Research Program of the National Institutes of Health (1ZIABC011383-14 to DRL). The contributions of the NIH author(s) are considered works of the United States Government. The findings and conclusions presented in this paper are those of the author(s) and do not necessarily reflect the views of the NIH or the U.S. Department of Health and Human Services. The funders had no role in study design, data collection and analysis, decision to publish, or preparation of the manuscript.

**Competing interests:** The authors have declared that no competing interests exist.

different loci in thousands of single cells, generating 'traces' that map the physical locations of identifiable loci across entire chromosomes [64]. Moreover, the method is often simultaneously extended to RNA, offering a crucial dataset to explore the relationship between transcription and chromatin organization [14,52,58,63]. Thus, chromatin tracing connects DNA conformation, transcription activity, and single-cell variability in a single assay.

However, localization error in fluorescence imaging sets a limit to accuracy which becomes uniquely challenging with chromatin tracing [63,67]. While there is an inherent uncertainty with all fluorescence imaging techniques, additional error stems from factors specific to the chromatin tracing approach [63]. For example, traditional denaturing chromosome tracing protocols utilizing high temperatures and high formamide concentrations, can perturb the natural chromatin by tens of nanometers [6], even despite recent improvements [6,7]. Still, reported errors range from 25-100 nm for chromatin tracing studies, which is frustratingly close to the distances of DNA-DNA interaction involved in gene regulation [10,30,52,63]. Thus, inaccuracies can significantly affect both the measurement of distances between loci and the biological interpretation, making accurate quantification and analysis difficult. For instance, even minor errors can obscure whether two loci are in a looped configuration [40, 72]. Additionally, localization error can artificially inflate the mean distance between fluorophores [23]. While correction methodologies exist [56,60], they often rely on assumptions about shape and variability or require additional experimental steps [25]. Furthermore, these approaches have only been used to investigate the distances between proteins within molecular complexes with relatively defined shapes, and hence the lack of variability within the assumptions of the approach. A comprehensive theory of error in chromatin tracing has not been developed.

One area where DNA tracing holds great promise is elucidating the necessary spatial interactions between enhancers and promoters [13,72]. Enhancers are short DNA sequences that lead to an increase in transcription activity despite being genomically distant from their target promoters [5]. A point of contention within the field is whether enhancer-promoter (E-P) distance is important for transcription regulation, and if it is, what timescales are needed for communication [12,70]. Models that assign importance to E-P proximity still predominate, but with diverse degrees of importance as to how E-P distance influences transcription [12,17,20,21,32,36,46, 52,66]. On the other hand, many studies suggest that E-P proximity is not correlated with transcription activation [3,55,57] or even that proximity decreases with transcription activity [2,9,44]. The complexity of the situation can be illustrated with the well-studied case of the *Sox2* gene and its super-enhancer (SE); chromatin tracing, live-cell imaging, and other chromatin capture methodologies have extensively probed this relation with various perturbations [3,19,24,38,57,65]. Overall, although *Sox2* transcription activity was generally shown to be related to E-P proximity at an ensemble level, no clear direct relation to nascent activity was found in individual cells (discussed more later). Further, other context dependent elements have now been found to play a role in the regulation of the *Sox2* gene [16,36], potentially suggesting that higher order multi-way interactions could play a role in its regulation [4,18,31,35,41, 46,66].

Here, we describe a methodology: Loci Enabled Advanced Resolution (LEAR), to improve the resolution of multi-loci microscopy. LEAR takes advantage of a unique feature of multi-loci microscopy data, which is the information that the loci are different and with known identities. The theory behind LEAR involves using an estimate of the localization error to calculate the true variance of the displacements between loci pairs, termed 'goal variances', and then iteratively adjusting localizations to match these variances. We validated LEAR using both empirical data, which showed significant reductions in localization errors, and realistic simulations, where contact frequencies previously distorted by errors were much more accurately determined. Applying LEAR, we explored how an inserted boundary impacts *Sox2* transcription, revealing that E-P proximity within individual cells influences nascent transcription, a relationship previously obscured by localization error. Further analyses suggest that chromatin capture methods could predict higher-order contact frequencies, with loop extrusion potentially serving as a switch-like mechanism to facilitate multi-way contacts.

## Design and implementation

### The theory of LEAR

There are now several technologies that break the Abbe resolution limit of microscopy [1]. All of these have different advantages and disadvantages in regard to the spatiotemporal-resolution available [27,61]. Naturally, these have led to additional post-processing methodologies to remove artifacts and to further improve resolution (a few examples: [15,28, 39,45,51]). Here we describe a post-processing methodology to improve the resolution of chromatin tracing capitalizing on the fact that there are many distinguishable loci: Loci Enabled Advanced Resolution (LEAR).

For one of the imaging dimensions with $L$ total loci, the observed locations of locus ($\ell$) for the $N$ traces is $O^\ell = (O_1^\ell, O_2^\ell, ... O_N^\ell)$. The observed locations result from the localization error, $\epsilon^\ell = (\epsilon_1^\ell, \epsilon_2^\ell, ... \epsilon_N^\ell)$, distorting the true positions, $T^\ell = (T_1^\ell, T_2^\ell, ... T_N^\ell)$, with $O^\ell = T^\ell + \epsilon^\ell$. Each element within $\epsilon^\ell$ follows a random variable whose specifics result from the particular experiment.

The overall goal is to obtain a better approximation of $T^\ell$, given the observed localizations of all loci: $\mathbf{O} = \{O^1, O^2, ... , O^L\}$. With the experimental data, we can empirically quantify the variance of the differences between a pair of loci ($\ell = \alpha$ and $\ell = \beta$); with the assumption that the localization errors are independent, the variance is the following:

$$Var(O^\alpha - O^\beta) = Var(T^\alpha + \epsilon^\alpha - O^\beta) = Var(T^\alpha - O^\beta) + Var(\epsilon^\alpha), \tag{1}$$

We now have an equation that allows us to approach $T^\alpha$. Put another way, if the localization error of locus $\alpha$ were minimized ($Var(\epsilon^\alpha) \to 0$) the variance above would approach $Var(T^\alpha - O^\beta)$; we refer to this quantity as the 'goal variance' which we can calculate given an approximation of $Var(\epsilon^\alpha)$. We describe a few approaches one can use to estimate $Var(\epsilon^\alpha)$ in specific cases in Sect 6 in S1 Text, but it is important to state that an estimation may need to be quantified experimentally.

The central idea is that we can minimize the localization error of locus $\alpha$ by adjusting $O^\alpha$ to produce a variance equal to the goal variance; again, where the localization error is minimized. More specifically, we can adjust the positions with the addition of $C^\alpha = (C_1^\alpha, C_2^\alpha, ... C_N^\alpha)$ generating the adjusted locations: $A^\alpha = O^\alpha + C^\alpha$, so that $Var(A^\alpha - O^\beta) = Var(T^\alpha - O^\beta)$. The thought being that $A^\alpha$ will better approximate $T^\alpha$ as $Var(A^\alpha - O^\beta)$ approaches $Var(T^\alpha - O^\beta)$ (Fig 1A).

To evaluate an $A^\alpha$ 'guess,' we can calculate the probability of observing a specific goal variance; this is done with the central limit theorem, resulting in a normal distribution with a mean ($\mu_{\alpha,\beta} \approx Var(T^\alpha - O^\beta)$) and standard deviation ($\sigma_{\alpha,\beta}$) for that pair of loci (Sect 2 in S1 Text).

$$P_{Var}^{goal}[Var(A^\alpha - O^\beta)|\mu_{\alpha,\beta}, \sigma_{\alpha,\beta}] = \frac{exp(-\frac{1}{2}(\frac{Var(A^\alpha - O^\beta) - \mu_{\alpha,\beta}}{\sigma_{\alpha,\beta}})^2)}{\sigma_{\alpha,\beta}\sqrt{2\pi}}. \tag{2}$$

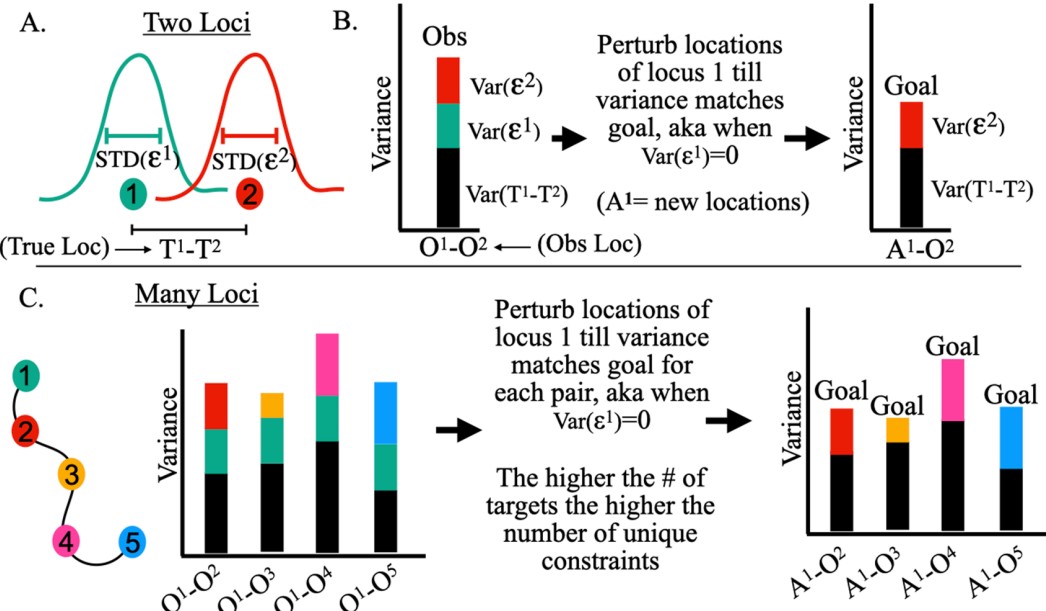

**Fig 1. Illustration of ideas behind LEAR:** A) There are two loci, each have true locations (illustrated as solid circles) and localization errors $\epsilon$ (illustrated with Gaussian like distributions and standard deviations). The true differences between the true locations is $T^1-T^2$. B) Calculating the variance between the localizations with the localization error (aka. the observed raw localizations) results in a variance made up of 3 terms owing to error propagation: one from the variance of the differences of the true locations and the other two equal to the actual variance of the localization errors (see equation 1). With an estimate of $STD(\epsilon^1)$, we can determine the goal variance; the variance if $STD(\epsilon^1) \to 0$. The main idea is that we can improve the resolution by adjusting the positions of $loci^1$ until the differences calculated with the adjusted positions ($A^1$) produce a variance equal to the goal variance. C) With a single pair of loci, there is only one goal variance, hence there are not many constraints. However, if there are many different loci, one can quantify a goal variance for each pair, drastically increasing the number of constraints to guide the method.

That is, we seek a conformation of $A^\alpha$ that maximizes the above probability density.

Problematically, with only one pair of loci, and hence one goal variance, many different $A^\alpha$ guesses will give the same goal variance. Importantly, however, with an increasing number of imaged loci, the number of goal variances increases, providing more constraints on $A^\alpha$ (Fig 1B). To incorporate these constraints, we can seek a conformation of $A^\alpha$ that maximizes the following expression, corresponding to a likelihood maximization problem:

$$l[A^\alpha] \propto \prod_{\ell}^{L} P_{Var}^{goal}[Var(A^\alpha - O^\ell)|\mu_{\alpha,\ell}, \sigma_{\alpha,\ell}]) \tag{3}$$

More clearly, we make the assumption that the likelihood ($l[A^\alpha]$) of having a "good" configuration is proportional to the product over all available loci $L$. The fact that the above equation is specific to a single locus allows one to parallelize the approach. To maximize the above equation, we utilized a simple stochastic descent algorithm (Sect 1 in S1 Text).

## Results

### Case 1: Demonstrating that LEAR improves localization error experimentally

We first wanted to test our methodology experimentally. To do this we utilized three previously published datasets where specific loci were repeatedly imaged within the same cell, allowing us to quantify the localization error by quantifying the distances between the initially imaged and the re-imaged localizations (Sect 5 in S1 Text). Two of the datasets were from

experiments within *Drosophila* from the work of Mateo et al. [52]; the first had a genomic resolution of 2kb and the second had a genomic resolution of 10kb. The third dataset was sequential chromatin tracing data within IMR90 human cells from the work of Su et al. [63] with a genomic resolution of 50kb. The number of imaged loci in each dataset varied as well as the number of loci that were repeatably imaged, with the 50kb resolution data having the most repeatably imaged loci $\approx 20$. These loci were repeatedly imaged in each cell specifically to quantify the localization error. The chromatin organizations of these datasets greatly varied, with the 2kb resolution data showing the smallest distances overall (Fig 2Ai,Bi and Ci), allowing us to demonstrate our methodology with diverse situations.

Here we pause to state that we filtered out some localizations that were reasoned to be "overwhelmingly" off target (discussed extensively within the Sect 8 in S1 Text, Figs A–D in S1 Text). Previous work has also filtered out localizations depending upon how the observed localizations deviated from a specific model [38,42]. However, little evidence was given to justify the discarding of certain localizations beyond a better agreement with chromosome capture contact frequencies [42]. Importantly, we provide multiple lines of evidence justifying 'filtering' within Sect 8 in S1 Text. A more complicated filtering approach rooted in polymer theory could be further developed taking into consideration the nature of off target localizations quantified here [42]. We must also state that due to this problem, the error improvement from LEAR is likely underestimated (Sect 8.2 in S1 Text). Throughout we only tested the application of LEAR on the initially imaged loci, for the repeat imaged loci showed a higher proportion of off target localizations (Figs C and D in S1 Text). Nonetheless, overall, we saw a general improvement with the application of LEAR no matter the dataset (Fig 2Aiii, Biii, and Ciii) as shown below.

We first quantified the amount of localization error for each of the datasets along each dimension; again, defined as the standard deviation of the localizations about their true localizations ($STD(\epsilon^{\ell})$). Using a simple process, we were able to determine the mean localization error for the initially imaged loci and the repeat imaged loci for the raw data (Sect 5 in S1 Text). The motivation for this came from the work of Su et al., where they noted a difference in the brightness of the repeat imaged localizations [63]—and hence, re-imaging the same locus in the same cell could be fundamentally different in terms of localization error. Our analysis supports this reasoning and quantifies the difference: we found that the repeat imaged localizations (generally) had a worse localization error (Fig 2Aii, Bii, and Cii). Of note, there is a drastic difference in the localization error for the *Z* dimension within the dataset of Su et al. On its own, this analysis is important regarding the actual resolution of an experiment—the localization error is apparently not accurately quantified using only the distances between initially imaged and repeat imaged loci. Simply put, for these chromatin tracing protocols, repeatably imaging the same locus can result in a different localization error than when the locus was initially imaged.

We then quantified the relative error of the initially imaged localizations after the application of our methodology. To do this we used the same logic as above to quantify the localization error but with the adjusted localizations of LEAR, and then quantified the relative localization error; *STD* with LEAR divided by the *STD* with raw localizations. We show the relative localization error for each loci for each dimension individually in Fig 2Aiii, Biii, and Ciii. Generally, we found that the larger the error and the smaller the distances between the loci, the more effective the error improvement from LEAR (discussed later). For the 2kb genomic resolution data, the *Z* dimension showed the greatest improvement ($\approx 20\%$), while the other dimensions showed an improvement of $\approx 10\%$ (Fig 2Aiii). The same trend was seen with the 10kb genomic resolution data, but with lower improvements as calculated with only two loci (Fig 2Biii). The dataset of Su et al. showed consistent small improvements throughout, again in line with the fact that a lower genomic resolution (larger distances between loci) decreases the effectiveness of the method. Altogether, these results show that LEAR works on experimental data improving the localization error by $\approx$ 10-20% for high genomic resolution chromatin tracing data.

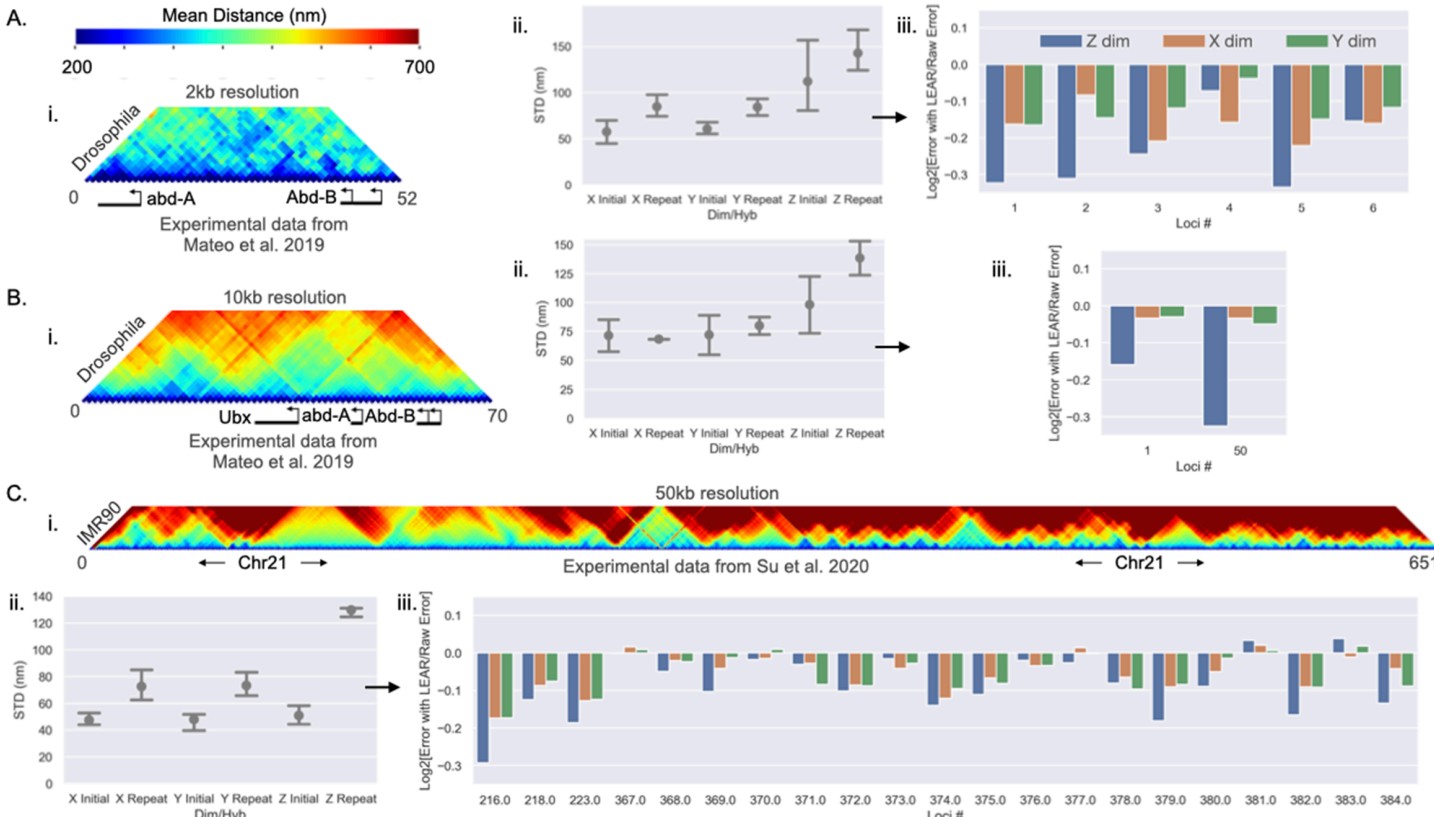

**Fig 2**. **Validating the methodology experimentally:** A, B and C) Are the analyses with the listed chromatin tracing dataset. i) The mean distances between the loci for the listed dataset. ii) The quantified localization error using the repeatably imaged loci from the dataset; the localization error is shown for each dimension and for whether it was from the initially imaged or the repeatably imaged. Error bars are equal to the standard deviation. iii) The quantified relative localization errors after the application of the methodology for the individual loci and along each dimension. Note, this was only done for the initially imaged loci.

## Case 2: Applying LEAR on the *Sox2* locus shows drastic improvements to contact-frequencies

Although we demonstrated LEAR using experimental data, the fact that the true underlying locations were unknown prevented us from determining how LEAR performed in relation to commonly used metrics like the contact-frequency. Additionally, it was likely that off-target localizations resulted in an underestimation of LEAR's error improvement with the experimental data, as described above. Therefore, we tested the methodology through simulations using realistic multi-loci microscopy data, where the ground truth was known.

To apply LEAR on realistic simulation data we utilized the empirical chromatin tracing data of Huang et al. [38], where they quantified the 3d positions of $\approx$ 40 loci distributed around the Sox2 gene at a genomic resolution of 5kb within mouse embryonic stem cells (mESC). The contact frequency quantified from $\approx$ 1400 traces with a distance threshold of 150 nm is shown in Fig 3A; loci lacking a sufficient number of datapoints were excluded. It is important to state that for chromatin tracing *contact frequency* refers to the proportion of traces where two or more genomic loci are spatially within a defined distance threshold (e.g., 150 nm or 200 nm), which may be different from the contact frequencies determined with other experimental methodologies. The idea was that the empirical data would be treated as the ground truth and different amounts of localization error would be computationally introduced, the performance of LEAR could then be evaluated with the knowledge of the true underlying locations with no off target effects. Because localization errors can lead to artificial

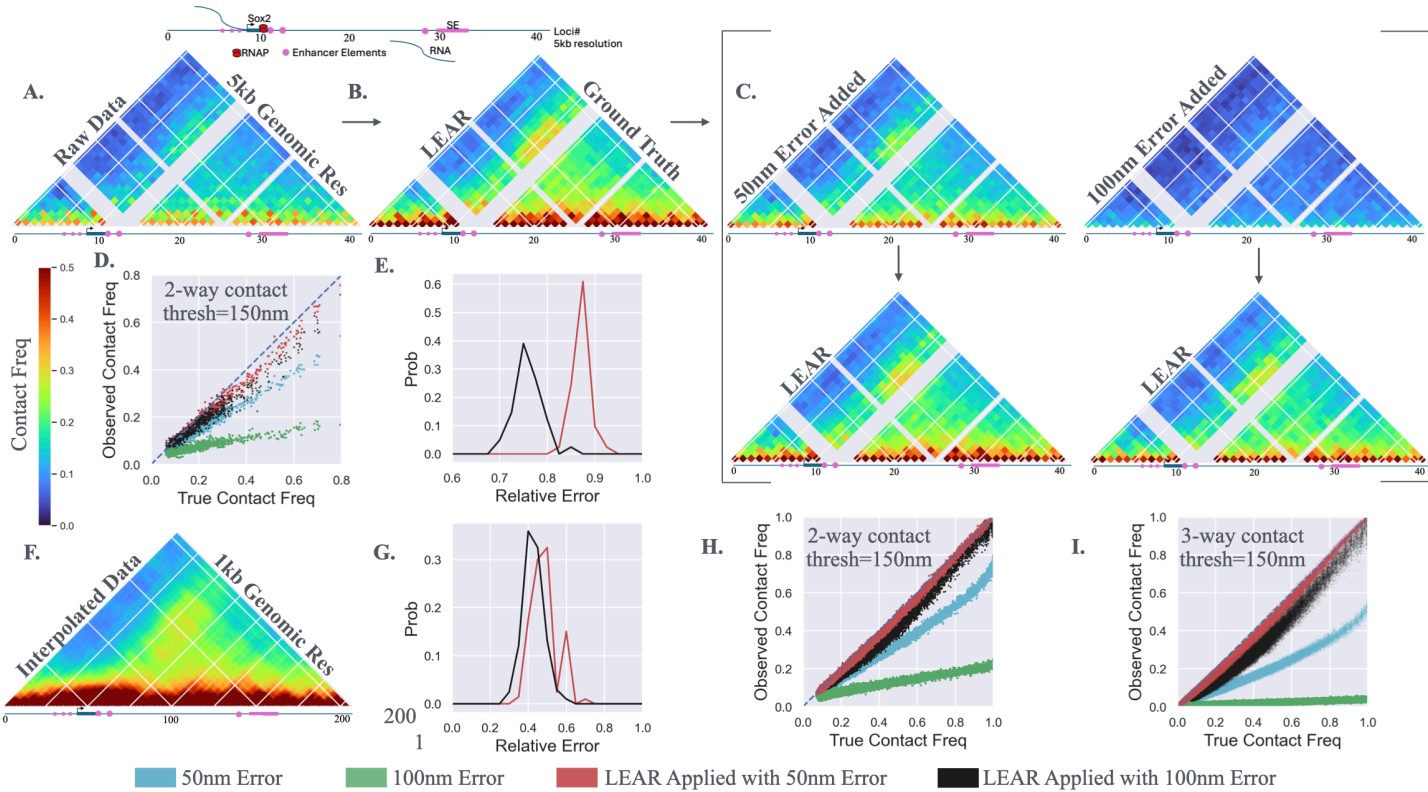

**Fig 3**. **Testing the Method with Simulation:** A) The contact frequencies using a distance threshold of 150 nm from the chromatin tracing data of Huang et al. [38]. B) The contact frequencies after the application of LEAR. This data was used as the ground truth. C) The contact frequencies after incorporating normally distributed localization error with standard deviations equal to $50nm$ or $100nm$. The arrows direct one to the contact frequencies that result from LEAR. D) The observed contact frequency vs. the true contact frequencies for the results shown in C—the blue line is for reference. Note, the colors for the conditions throughout are shown at the bottom of the figure. E) The average relative localization error after applying the methodology. F) A linear interpolation of the data in B, approximating a dataset with a genomic resolution of 1kb. G) The relative error for the interpolated dataset with the two localization errors. H) Similar to D but for the interpolated 1kb genomic resolution dataset. I) The observed 3-way contact frequency vs the true 3-way contact frequency for the interpolated dataset (how often three different loci are within 150 nm of each other).

increases in mean distances [23], we first applied LEAR utilizing estimates of the localization error within the Huang et al. dataset (Fig F in S1 Text) to generate a more "realistic" dataset, especially in terms of ensemble measurements (Fig 3B, Sect 7 in S1 Text, validated throughout). This dataset was used as the ground truth for the following simulation experiments within the main text (Fig 3). We also performed the same analysis using the raw localizations as the ground truth (Fig 3A) and observed similar results but with slightly less error improvement due to the larger distances between the loci (Fig G in S1 Text, discussed later).

We quantified the efficacy of LEAR by introducing normally distributed localization error and then testing how well we recovered the 'ground truth' post LEAR processing. We added the localization error using normally distributed random numbers with either $STD(\epsilon^{\ell}) = 50nm$ or $STD(\epsilon^{\ell}) = 100nm$ (Fig 3C, $STD$ is the standard deviation). As expected, the contact frequencies were greatly perturbed with the 100 nm error showing the largest reductions. The application of the methodology resulted in contact frequency maps that nicely approximated the ground truth (Fig 3C, below arrows)— illustrating a clear benefit of the LEAR methodology. To show these results more clearly, we plotted the observed contact frequency vs. the true contact frequency for the different conditions (Fig 3D), where there was a clear improvement for

both the 50 nm and 100 nm error, especially for the larger contact frequencies. To measure the improvement at the single molecule level we quantified the average error using the distances between each localization and their true localizations, and then calculated the relative error of LEAR; similar to before, the relative error is the average error with LEAR divided by the average error before the application of the method. This analysis showed that the 100 nm localization error had about 25% less error while the 50 nm error showed a lesser improvement of $\approx 12\%$ (Fig 3E). We also show a plot of $C$ vs. $\epsilon$ to further illustrate that LEAR works at the single molecule level (Fig H in S1 Text). The analysis suggests that the degree of improvement is sensitive to certain aspects of the data. Nevertheless, qualitatively one sees the off diagonal peak in Sox2 contact with the super enhancer with greater contrast after LEAR processing.

We then sought to probe what aspects of the chromatin tracing data influences performance. Most importantly, we hypothesized that the effectiveness of the methodology is proportional to the size of the error ($Var(\epsilon^\alpha)$) and inversely proportional to the variance of the differences ($Var(O^\alpha - O^\beta)$) (see Sect 3 in S1 Text); generally, the larger the distances between neighboring loci the worse the performance. To test this hypothesis, we compared the above results to computational experiments that utilized the actual raw localizations as the ground truth instead (Fig G in S1 Text)—the raw localizations have larger mean distances, and naturally, larger variances from the localization error. The improvement in the quantification of contact-frequency remained noticeable showing significant improvements (Fig Gd in S1 Text). However, and as expected, we observed less relative error improvement with $\approx 15\%$ for the 100 nm localization error and $\approx 8\%$ for the 50 nm (Fig Ge in S1 Text). For completeness, we then applied our methodology on chromatin tracing data collected at a genomic resolution of 30kb [10] which showed consistent results with this line of reasoning (Fig I in S1 Text). Note, we also investigated aspects of the methodology and chromatin tracing data that could affect the performance of the methodology Fig Ib–e in S1 Text, as discussed extensively discussed within the Sect 4 in S1 Text. Together, these results clearly explain the differences in performance for the diverse experimental datasets (Fig 2Aiii, Biii, and Ciii), and for datasets similar to that of the *Sox2* data here, improvements in both contact-frequency calculations and resolution can be substantial.

We next sought to understand how the methodology performs in certain limiting conditions expected to deliver large improvements in resolution post-LEAR. Such conditions would be chromosomes where the number of imaged loci is large and the distances between neighboring loci is small. To computationally generate such a dataset, we performed a linear interpolation of the ground truth data (Fig 3B), generating interpolated chromatin tracing data at a genomic resolution of 1 kb (Fig 3F). As before, we introduced the two localization errors, applied the methodology, and found large improvements in the relative error for both the 50 nm and 100 nm (Fig 3G), with $\approx 50\%$ improvement for both cases. And, in terms of 2-way contact-frequency, we found that the methodology was able to obtain the correct result no matter the true underlying contact frequency (Fig 3H). Lastly, to further highlight the effectiveness of LEAR, we decided to investigate 3-way contact frequencies (Fig 3I); that is, how often three different loci are within a certain distance of each other. The detrimental effect of the 100 nm localization error is shown in green in Fig 3I. Nicely, the methodology allowed us to obtain dramatically improved 3-way contact frequencies—clearly showing the benefit of LEAR.

In summary, this analysis demonstrates that LEAR greatly reduces the error in quantifying contact-frequencies and as the genomic resolution of the chromatin tracing data increases so does the benefit of applying LEAR. The improvement in quantifying contact-frequencies was especially apparent for multi-way contacts. Our investigations with the *Sox2* data suggest that applying LEAR to similar datasets improves localization error by approximately 10-20%. And lastly, with future technological advances in chromatin tracing, where more loci are analyzed at even finer resolutions, our idealized system experiments indicate that LEAR could potentially reduce localization errors by up to 50%.

## Case 3: Quantifying the impact of LEAR with polymer simulations

Next, we sought to apply and test LEAR utilizing polymer simulation data as the ground truth. The application of LEAR in Case 1 and Case2 was motivated from experimental data [38,52,63], indicating that LEAR will prove beneficial in a real

life scenario. However, because of the potential experimental errors the use of these datasets as a ground truth for the chromatin structure may not represent the true underlying structure.

We decided to test LEAR utilizing the recent polymer simulation data of Jusuf et al. [43]. Though the 1 kb genomic resolution polymer simulation was originally performed at a chromosomal scale with several CTCF sites and loop extrusion, we decided to focus on the first 100 loci of the dataset with 500 individual traces within the main text (Fig 4A). We also analyzed a region with CTCF sites with similar results in S1 Text (Fig J in S1 Text). Because of the equal large improvements with both the 50 nm and 100 nm localization errors obtained with the interpolated 1 kb dataset (Fig 3G), we decided to further test the limits of LEAR by introducing normally distributed localization error with either $STD(\epsilon^\ell) = 25nm$ or $STD(\epsilon^\ell) = 50nm$. We hypothesized that the improvements seen with LEAR would be less for the smaller error, as discussed above.

We found that LEAR performed similarly when compared to the interpolated 1kb dataset (Fig 3G). As before, LEAR was able to correctly determine 2-way and 3-way contact frequencies with both types of error (Fig 4B and 4C). Importantly, we observed that the relative localization error per locus still showed significant improvements for the 25 nm localization error, with about a 35% improvement (Fig 4D). The relative improvement for the 50 nm localization error was around 50%, as in the interpolated 1 kb data in Case 2. To make it more clear that LEAR is able to improve the localization error at the individual trace level, we quantified the relative localization error per trace ($1/N \times \sum_{\ell=1}^{N}(A_i^\ell - T_i^\ell)/(O_i^\ell - T_i^\ell)$, for each trace $i$) . As expected we saw similar improvements for the two localization errors.

Overall, Case 3 demonstrates that LEAR performs similarly on polymer simulation datasets. Of note, we also found that LEAR can lead to error improvements $\approx 35\%$ for even small localization errors of $STD(\epsilon^\ell) = 25nm$, given the fine genomic resolution of the simulation. Together these further support the potential use of LEAR in diverse experimental situations.

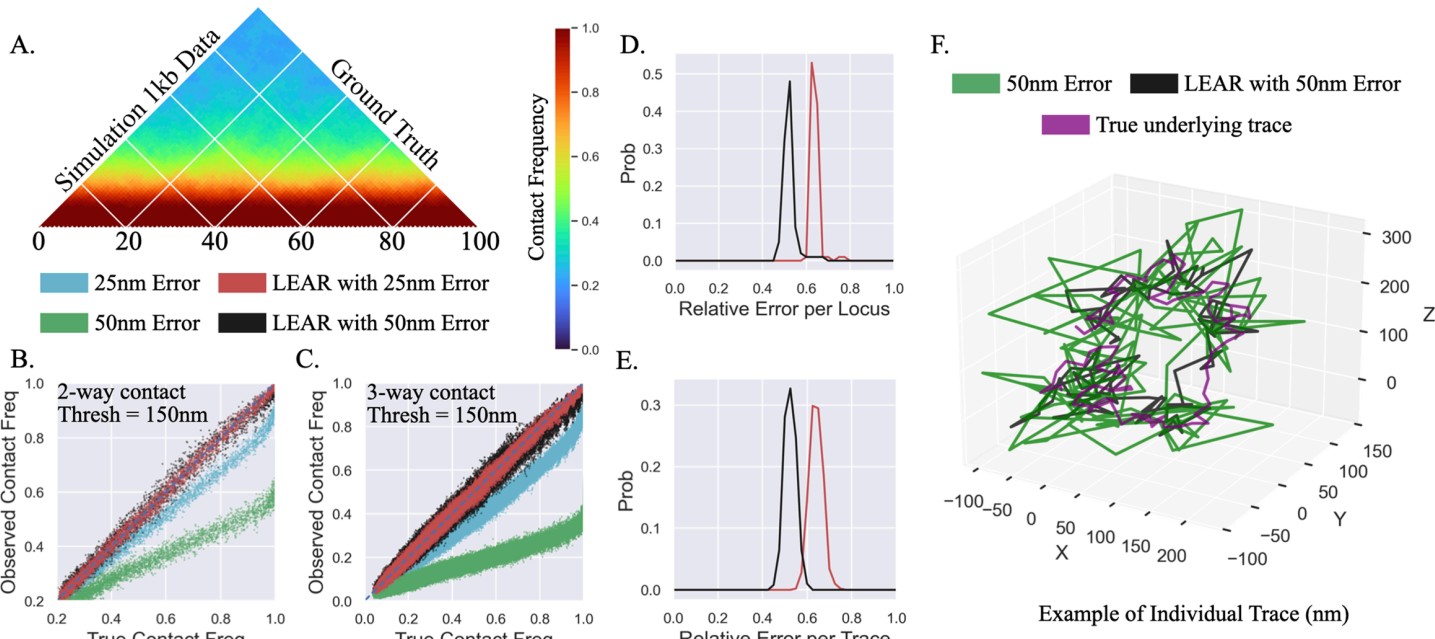

**Fig 4**. **Validating the methodology with polymer simulations:** A) The ground truth contact frequency map of the first 100 loci of the 1 kb polymer simulation data [43]. B) The observed 2-way contact frequencies vs. the true underlying contact frequencies for 25 nm or 50 nm localization error both with and without LEAR—again blue line is for reference. C) The same as in B but for all possible 3-way contact frequencies. D) The relative localization error for each individual locus after the application of LEAR for the 25 nm and 50 nm errors. E) The relative localization error for each individual trace of the polymer simulation data. F) Example of individual trace with underlying ground truth in purple, LEAR in black and with localization error in green.

## Transcriptional bursting of *Sox2* is related to E-P proximity in individual cells

The *Sox2* gene and its super-enhancer (SE) represents one of the most investigated enhancer-promoter (E-P) pairs. However, the role of 3D chromatin organization in regulating Sox2 transcription remains unclear [3,16,19,24,38,57,65]. Although ensemble data shows a correlation between the contact frequency of the super-enhancer and *Sox2* transcription [19,24,38], no clear relationship has been established between individual transcriptional bursts and the distance between the super-enhancer and promoter in single cells [3,38,57], raising questions about the relevance of E-P spatial proximity and transcription regulation. Specifically, live-cell microscopy has shown no relation between E-P distance and nascent transcription [3,57]. However, chromatin tracing has resulted in conflicting results [38]; a weak but significant change in the median E-P distance with nascent transcription was observed (a change of $\approx$ 18 nm), but there was no clear difference in the proportion of alleles showing active transcription as a function of E-P distance threshold. We therefore sought to use the improved localization error of LEAR to investigate the relation between *Sox2* transcription and local chromatin organization with the chromatin tracing data of Huang et al. [38]—the dataset from Fig 3.

We first wanted to investigate the contact frequencies of the *Sox2* locus with LEAR with a focus on the effect of a strong boundary element between the SE and the promoter. Previously, Huang et al. reported that insertion of multiple CTCF sites led to a reduction in the SE-P contact frequency and a decrease in the fraction of transcriptionally active *Sox2* alleles [38]. Using the raw chromatin tracing data we show the contact frequencies (thresh = 200 nm) for the no boundary *Sox2* allele, with the boundary, and then the difference between the two conditions to emphasize the change (Fig 5B, red indicates a loss of contact frequency, top row). As in Huang et al., we observed a loss in contact frequency between the promoter and the SE region (Fig 5B, top right), with an absolute change in contact frequency of $\approx$ .07. After applying LEAR, the contact frequencies dramatically increased for both conditions, and the absolute difference in chromatin structure was much greater (Fig 5B, bottom row). This indicates the effect of the boundary insertion on chromatin organization was more substantial than previously measured, suggesting that the observed transcriptional changes might be linked to broader conformational shifts that were previously masked by localization errors.

We then aimed to assess how chromatin structure changes with nascent *Sox2* transcription at the single-cell level. To do this, we compared contact frequency maps when nascent RNA was detected versus when it was not. In Huang et al.'s study, a fluorescent protein (FP) was inserted directly after *Sox2*, allowing the RNA-FISH to differentiate between nascent RNA containing both FP and *Sox2* signals ('later burst detection') and only *Sox2* signal ('initial burst detection'); the lack of FP signal indicates that RNA polymerase did not have enough elongation time to reach the FP sequence (illustrated in Fig Ka in S1 Text). For our analysis, we decided to only use traces with initial burst detection for the traces with active transcription, chromatin diffusion during transcription elongation distorts the E-P distances associated with transcription initiation [14]. Note, we also performed the same analysis without excluding traces where we observed a similar but lesser correlation between SE-P proximity and nascent transcription (Fig K in S1 Text). For the no boundary condition contact frequency maps and the difference between them, we observed some changes focused mostly around the promoter region, with one of the more notable changes being located at one of the recently characterized enhancer like element [16] (Fig 5C, top row). After applying LEAR, the contact frequencies increased, and the changes were slightly more noticeable (Fig 5C, bottom row). Though this result may suggest some correlation of chromatin structure and transcription regulation, the contact frequencies of the P-SE showed no real variation with nascent transcription. For the boundary condition (Fig 5D, bottom row), we observed a change centered around the inserted boundary element with transcription (Fig 5D, top row), which is an interesting result when considering the recent models of CTCF hubs enabling E-P regulation within different chromatin domains [35,41]. Notably, with LEAR, we found a noticeable change centered around the SE-P contact frequencies, indicating that with the insertion of the boundary element, the spatial proximity of the SE may be important for the initiation of *Sox2* transcription. Importantly, without LEAR, this relation would have likely been missed.

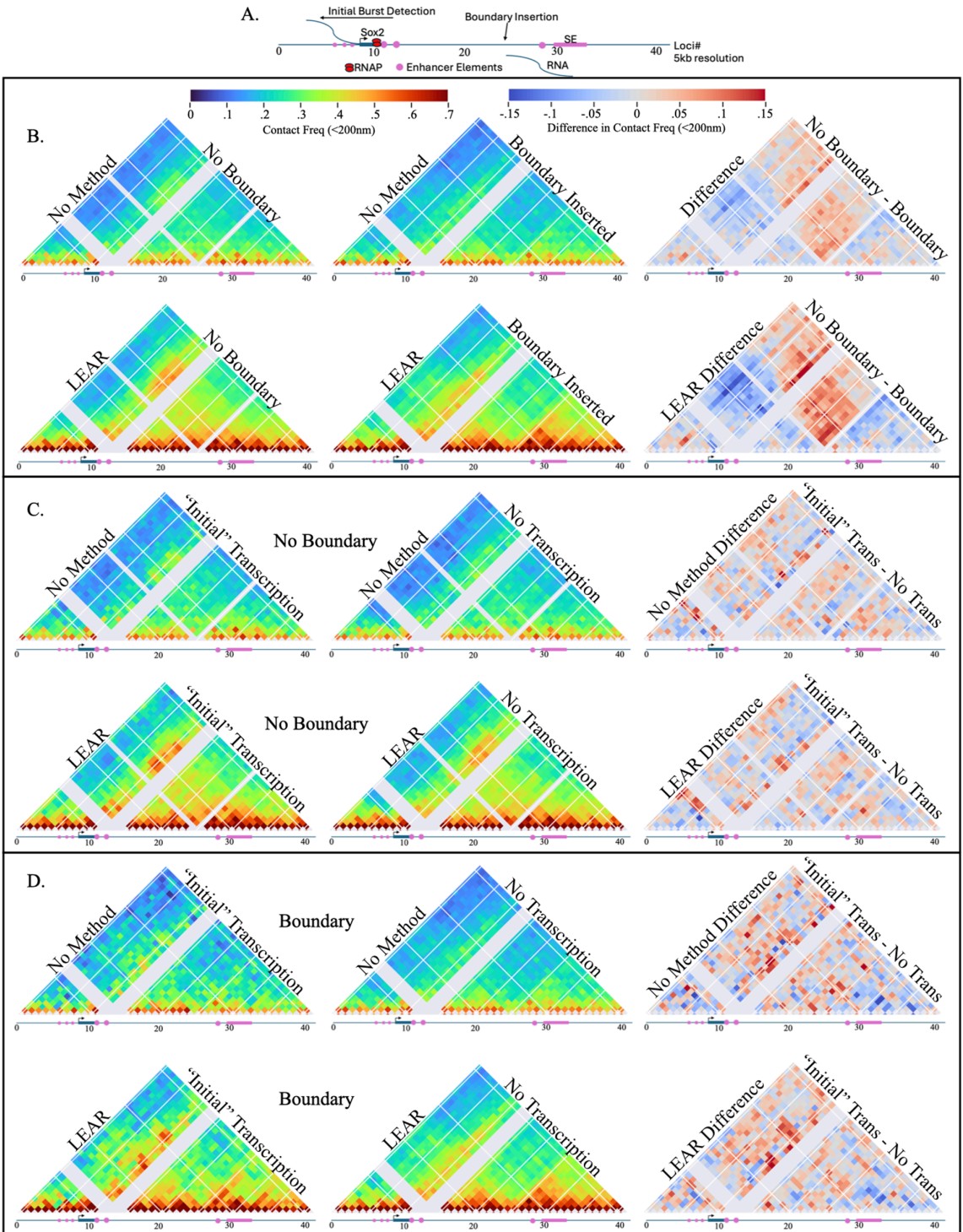

**Fig 5. Link between Sox2 transcription and local chromatin organization:** A) An illustration showing important factors around the *Sox2* gene, also shows the indices used for the 5kb chromatin tracing experiments [38]. B) The contact frequencies for with and without the boundary inserted, and also with and without the methodology being applied (see labels). The right column shows the difference in contact frequency for the specific row. C) Contact frequency maps for the no boundary system with traces where an initial transcriptional burst was seen (left column) and from traces where no transcription was observed (center). The difference of the contact frequency maps with and without initial transcription is shown on the far right. The contact frequency maps using the raw data are shown on the top row (and labelled), the maps generated utilizing the method are on the bottom row. D) The same as *C* but for the boundary experimental dataset.

Overall, we have demonstrated the usefulness of the methodology investigating the link between chromatin organization and *Sox2* transcription regulation. Not only did we show that changes in chromosome contacts were greater than previously quantified, we also found a link between chromatin organization and individual *Sox2* transcriptional bursts previously obscured by error.

## Cooperativity in multi-way contacts is not explained solely by loop extrusion

Emerging evidence suggests that multiple enhancers work together as a system, necessitating accurate and sensitive methods for detecting multiway contacts for understanding enhancer biology [12,66,68]. Considering that multi-way contacts are difficult to detect by chromosome conformation capture and were especially sensitive to localization error (Fig 3I), we sought to utilize the LEAR methodology to probe higher order contacts vis-a-vis loop extrusion. Previously, the work of Bintu et al., obtained rich chromatin tracing datasets in HCT116 cells with and without loop extrusion [10], which was eliminated with the degradation of RAD21 using the auxin degron system [59]. For the Bintu et al. dataset, we found a sufficient number of traces with a detection efficiency above 95% (7591 traces without auxin and 2726 traces with auxin), allowing for a deep interrogation of higher order contacts with the application of our methodology.

We conducted an in-depth analysis of loop extrusion on two- and three-way contacts. To start, we quantified the effect of LEAR on two way contacts with and without loop extrusion (Sect 9 in S1 Text). The 2-way contacts with a distance threshold of 200 nm are shown in Fig 6A, without (top row) and with the application of LEAR. To more clearly show the effect of loop extrusion on 2-way contact frequencies, we plotted the frequency without auxin vs. with auxin (Fig 6B). Though LEAR generally showed greater differences in 2-way contact frequencies due to loop extrusion, the results were mainly the same. We found that larger fold changes (not due to error) were around a $\approx .025$ contact frequency for loci pairs in the no loop extrusion condition, in some cases approaching a ten-fold increase with loop extrusion (See zoom in for Fig 6B).

We next sought to investigate higher order multi-way contacts with and without loop extrusion. To do this, we first quantified the proportion of traces where all three loci were within 200 nm and compared the expected 3-way contact frequency; calculated assuming the 2-way contact frequencies contributed independently (Sect 10 in S1 Text). We found a strong enrichment above the expected independent 3-way contact frequency indicative of cooperativity (Fig 6C and 6D). While cooperativity in multi-way contacts has been seen in cells lacking loop extrusion, a careful quantification of the relation between loop extrusion and cooperativity was lacking [10]. Astonishingly, we found that the auxin and no auxin condition followed the same curve both without (Fig 6C) and with LEAR (Fig 6D). Probing further, we performed the same analysis but for 4-way contact frequencies and obtained a similar result (Fig L in S1 Text). These outcomes are significant because it indicates that the cooperative behavior primarily hinges on the frequencies of two-way contact frequencies. Put another way, the underlying mechanisms influencing individual two-way interactions do not alter the degree of cooperativity, suggesting loop extrusion is not the mechanism behind the cooperative behavior and that the true underlying mechanism is mainly dependent upon spatial proximity.

We next attempted to more carefully quantify how the higher order multi-way contacts vary with loop extrusion but from a different perspective. To do this, we plotted the multi-way contact frequency with loop extrusion vs. without loop extrusion, for both 3-way (Fig 6E) and 4-way (Fig 6G) contact frequencies. We observed a large range of values with LEAR, approaching a max of .5 and .3 for the 3-way and 4-way contacts respectively, suggesting that even though the distances between pairs of loci show a high degree of variability [29], a large proportion of traces exhibit higher order multi-way contacts. To highlight the difference in multi-way contact frequency with LEAR, we quantified the difference in multi-way contact frequency due to LEAR and plotted it vs the multi-way contact frequencies with auxin (Fig 6F and 6H). We found that the change due to LEAR was drastic, again showing just how sensitive these multi-way contacts are to localization error and demonstrating the need of the method.

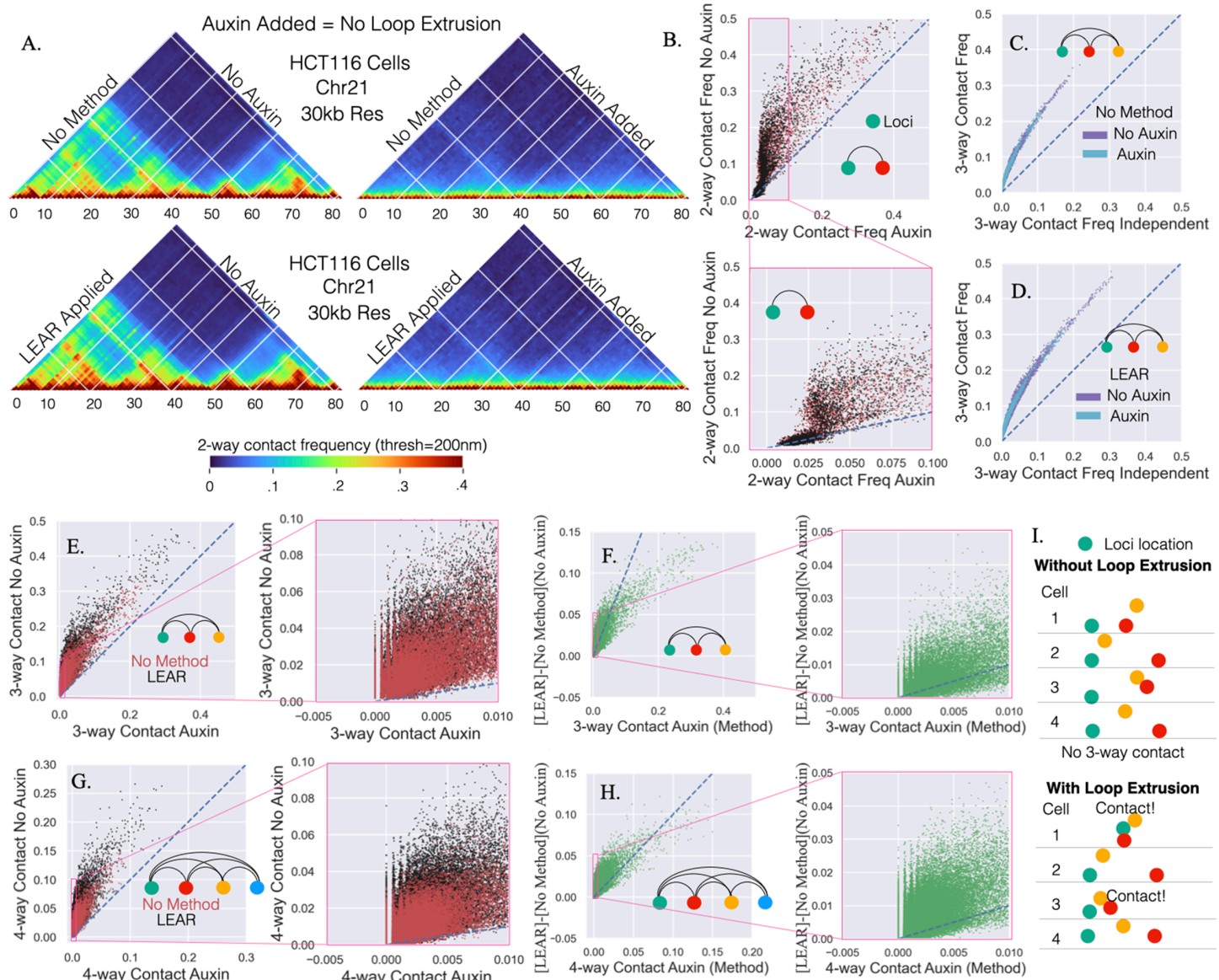

**Fig 6. Higher order contacts with and without loop extrusion:** A) The 2-way contact-frequencies for the chromatin tracing data from Bintu et al. The addition of auxin leads to the removal of active cohesion eliminating loop extrusion. The same chromosomal region is shown with and without the methodology, and with and without the addition of auxin. B) The 2-way contact-frequency without auxin vs. that with auxin, with and without the application of the methodology, and below is a zoom in of the top plot. C) The 3-way contacts between triplets compared to the expected 3-way contact frequency if the 2-way contact frequencies contributed independently, with and without auxin. D) The same as *C* but with the methodology. E) Same as *B* but with 3-way contact frequencies. F) The difference between the method and the no method 3-way contacts for the no auxin condition vs the 3-way contact with auxin. G) The same as *E* but with 4-way contact frequencies. H) Same as *F* but with 4-way contact frequencies. I) An illustration pointing out that for certain 3-way contacts the loci are never in contact without loop extrusion, but with loop extrusion a notable number of 3-way contacts are seen.

Lastly, we focused our attention on frequencies that showed the most dramatic changes with loop extrusion (Fig 6E and 6G, inset). Interestingly, we observed that initially zero multi-way contact frequencies without loop extrusion could increase to ≈ .04 with loop extrusion (Fig 6E and 6G, inset). This result implies > 40-fold changes, considering we only

quantified multi-way contact frequencies if there were at least 1000 distances between all pairs of loci (Fig 6E and 6G, zoom in). In other words, the analysis suggests that without loop extrusion certain multi-way contacts 'never' happen, but with loop-extrusion the frequency can dramatically increase (Fig 6I).

In summary, we have performed an in-depth analysis of cooperative multi-way contact frequencies with LEAR. We find that multi-way contacts were greatly perturbed by localization error and become more prevalent with the use of LEAR. We also found that loop-extrusion, while clearly altering 2 way contact frequencies, was not the mechanism leading to high-order cooperativity. Rather, we found that the changes in 2-way contact frequencies brought about by loop-extrusion could lead to drastic many-fold changes in multi-way contact frequencies. We propose this result for multi-way contact frequencies arises from the cooperativity mechanism being primarily a function of spatial proximity.

## Discussion

Here we develop LEAR, a methodology to improve the resolution of multi-loci microscopy data. A central advantage is that the method is a post-processing methodology that does not require additional experimentation. We developed the theory and thoroughly verified the approach using both realistic simulations and experiment. The effectiveness of the approach was proportional to the 'size' of the error and inversely proportional to the distances (nm) between loci— meaning that the smaller the distances one wishes to measure, the better the improvement. With future advances in experimental techniques, we estimate that the localization error can be reduced by 50% within individual cells in ideal conditions. Contact frequency quantification was found to be highly sensitive to localization error, often leading to significant underestimation of true frequencies–particularly for higher-order multi-contact interactions. This sensitivity poses challenges for validation, which often relies on consistency between ensemble chromatin capture and chromatin tracing contact frequencies, suggesting that prior comparisons may warrant reexamination using this method. We then utilized the methodology to probe enhancer-promoter distance with respect to transcriptional bursting and found that indeed spatial proximity between these two elements was correlated with RNA synthesis, partially resolving previous discrepancies at the *Sox* locus. Furthermore, we used LEAR to investigate cooperative multi-way contact frequencies, where a central finding was that loop-extrusion is not the mechanism driving cooperatively.

With the application of LEAR, our analysis indicates a correlation between *Sox2* nascent transcription and SE-P proximity in individual cells that was previously occluded by localization error (Fig 5D). While *Sox2* transcription was generally shown to be correlated with E-P proximity at an ensemble level, no clear correlation to nascent transcription was found at the level of individual cells [3,19,24,38,57,65]. Our revised analysis and subsequent finding is particularly significant given that the *Sox2* system is a prominent example where no correlation between single-cell SE-P proximity and nascent transcription was previously observed with live cell microscopy [3,57]. Our results do not contradict prior studies; rather, the correlation we observed was only detectable upon the insertion of a boundary element between the Sox2 gene and its associated SE. In terms of mechanism, it is tempting to consider the role of other enhancer-like elements thought to contribute to *Sox2* activation [16] and DNA sequences immediately downstream of the *Sox2* gene that are critical for maintaining transcription independent of boundary elements [36]. The mechanism we currently favor is related to the observation that CTCF hubs tend to form toward the center of surrounding chromatin domains. This phenomenon can bring enhancers from different contact domains into proximity with their target promoters [35,41]. Intriguingly, we found that the Sox2 gene was more spatially proximal to both the inserted boundary element and the SE when transcriptionally active, suggesting that this mechanism may partially explain our findings.

Several sources of error likely still obscure the correlation between enhancer-promoter (E-P) proximity and nascent transcription. Although we enriched our dataset for traces temporally closer to transcription initiation, the fixed-cell nature of chromatin tracing limits temporal resolution, potentially masking underlying relationships [14]. Fixation protocols, similar to those observed in traditional denaturing chromosome tracing methods can disrupt natural chromatin organization [6,7,62]. Moreover, accurately assigning localizations to the correct trace remains challenging, and misassignment would

clearly distort quantified E-P relationships [42]. For instance, extra localizations in the same genomic region–arising from sister chromatids, aneuploidy, or imaging noise–complicate trace assignment [42]. Finally, the uneven stochastic nature of probes labelling a specific chromatin region introduces even more uncertainty, and the field has yet to address the fact that a localized segment of chromatin is not truly a fluorescent point source. Thus, just as LEAR enhanced the correlation between E-P proximity and nascent transcription, addressing these other error sources will likely further resolve discrepancies.

Lastly, one striking observation in this study is that the application of the methodology dramatically increased multi-way contact frequencies. Therefore, if an absolute relationship between these multi-way contacts and some biological phenomenon exists [12,47], the use of the method is vital. While evidence does seem to suggest that these multi-way contacts may be important for the formation of enhancer networks and transcription regulation, we must emphasize that the field is just beginning to quantify these relationships [12]. Interestingly, due to cooperativity, higher-order multi-way contacts could be essentially absent following the elimination of loop extrusion, exhibiting more of an "on/off" mechanism. In contrast, while 2-way contacts decreased with the elimination of loop extrusion, they were never entirely absent due to the stochastic nature of the chromatin polymer [29]. Importantly, we found that the mechanism underlying the cooperativity was not loop extrusion, supported by the observation that multi-way contact frequencies depended solely on the underlying pairwise contact frequencies. That is, the relationship between 2-way contact frequencies and higher-order contacts remained consistent regardless of whether loop extrusion was eliminated. A plausible explanation for such cooperativity is global variability across the region. For example, shifts in nuclear positioning could affect the overall compaction of the region as a whole and naturally produce elevated 3-way contact probabilities beyond those expected from independent pairwise interactions. However, this mechanism does not currently explain why the cooperative behavior was solely dependent upon the underlying pairwise contact frequencies, and if it is not a result of global changes, it indicates that the underlying mechanism is more dependent upon chromatin affinities. Lastly, if the trend holds, it suggests that chromosomal capture contact frequencies could potentially be used to predict higher-order contacts using an accurate lookup table. More generally, these observations provide physical insight into how the effects of enhancers which can often be subtle at the individual level could cooperate to give large changes in transcription and hence cellular physiology.

## Availability and future directions

The application of advanced microscopy methods will undoubtedly continue to push the resolution limits of chromatin tracing, and LEAR will still have utility in certain contexts. Even with the advances of non-denaturing chromatin tracing [6,7], localization error still stems from two central sources: (1) a limited number of photons and (2) the fact that a localized segment of chromatin is not truly a point source. While the first source may be overcome to an extent by advanced imaging technologies like MINFLUX [34], at current, it is not so obvious how the second will be overcome. The improved resolution from advanced techniques will surely limit the utility of LEAR, due to the improvement being proportional to the 'size' of the error. Still, given the limited availability and experimental disadvantages of advanced imaging approaches, we believe the use of LEAR will still find traction within the field—especially as research pursues more fine genomic resolutions. The idea behind LEAR could be extended to problems other than those of chromatin tracing experiments. The theory should be applicable to situations where one seeks an improvement of the positions of many identifiable targets (loci) in many different repeats (traces), for example investigating RNA structure utilizing multiplexed RNA FISH data [26,71]. We also believe that the theory and implementation could be further adjusted likely leading to additional improvements. An interesting direction would be incorporating the number of photons detected for each individual localization into the theory or the use of other machine learning techniques to better capitalize upon this previously overlooked information. The code for LEAR can be found here: https://github.com/CHB-Bohrer/.

## Supporting information

**S1 Text Figures:**

**Fig A**: **Illustration of 'displacement error' and 'minimum distance to neighboring loci':** On the far left is an example trace with the initially imaged loci in black and those that were re-imaged in red. The index of the loci are explained within the figure. The center two boxes illustrate the definition of 'Min dis to neigh loci' using either the initial imaged loci (bottom box) or the re-imaged loci (top box). The boxes illustrate that the distances to the neighboring loci are calculated and then the smaller one is taken; using either the initially imaged or the re-imaged. Each dot on the plot on the bottom right is from a single trace. The color of each dot shows the displacement error, which is calculated as the distance between the initial and re-imaged localizations for each trace. The 'displacement error' is essentially a proxy for the localization error of individual localizations, and the calculation is illustrated with the top right box. In the plot we also highlight three regions of interest (ROI), and each of these are described within the text of S1 Text.

**Fig B**: **Investigating Su et al. chromatin tracing data and localization error:** A) The probability distributions for the displacement error (the distance between initially imaged loci and re-imaged loci) for each of the imaging dimensions; blue is $Z$ dimensions and green and yellow are $X$ and $Y$. We also show with simulation the probability distribution for loci with a standard deviation equal to 100 nm; black dashed line. B) The same as 'A' but with a log scale to highlight the long tail of the distributions. C) The minimum distance to a neighboring loci for the re-imaged localization vs. the minimum distance to a neighboring loci for the initially imaged localization, where the color shows the displacement error along the labelled dimension. Please note the metrics are illustrated in Fig A. D) The cumulative distribution for all minimum distances to neighboring loci for each of the individual dimensions, highlighting that a small proportion of localizations have high minimum distances to neighboring loci. E) The cumulative distribution for the detection efficiency of individual traces. The dashed line is for the traces before eliminating localizations with high minimum distances to neighboring loci while the solid line is after.

**Fig C**: **Investigating 2 kb chromatin tracing data and localization error of Mateo et al:** The same analysis as in Fig B but with the 2 kb chromatin tracing data of Mateo et al.

**Fig D**: **Investigating 10 kb chromatin tracing data and localization error of Mateo et al:** The same analysis as in Fig B but with the 10 kb chromatin tracing data of Mateo et al. Note, here the number of loci that were re-imaged was two, which is why the amount of data is much less.

**Fig E**: **Comparing multiple chromatin tracing datasets suggests that localizations with high minimum distances to neighboring loci are off target:** A, C, and E) The mean distances between loci for the specified chromatin tracing dataset, in E we also highlight the approximate genomic locations of the maps shown in A and C. B, D and F) The cumulative distribution function for the different chromatin tracing datasets with zoom ins highlighting the largest minimum distances to neighboring loci. Note that for 'B' and 'D' we artificially made the genomic resolution 90 kb so that they would be more comparable with 'F.' That is, the larger the genomic resolution the larger the distances between neighboring loci.

**Fig F**: **Approximating the localization error of Huang et al.:** A) The 'worst case scenario' quantified for the different dimensions of the raw dataset. B) The cumulative distribution for the distances between neighboring loci at different genomic resolutions. These distances were only quantified along the $X$ dimension.

**Fig G**: **Testing the Methodology with Simulation Raw Data**. These results are similar to Main Fig 2. A) The contact frequencies using a distance threshold of 150 nm for the chromatin tracing data of Huang et al. [38] (no boundary condition). B) We do not apply our methodology and just computationally add the localization error. That is, we use the raw empirical locations as a ground truth. C) We add localization error directly to the raw data and generate the contact frequencies for the labeled localization error. In the lower row the contact frequencies that result from the application of the methodology

are shown. D) The observed/quantified contact frequencies vs. the true underlying contact frequencies, again quantified using the raw data. E) The relative amount of localization error quantified for individual cells, see main text.

**Fig H**: **Further Justification for LEAR at the Single Molecule Level**. The relationship of the correction terms ($C$) generated by LEAR to create the adjusted locations vs the error term added to the underlying ground truth ($\epsilon$). The black line highlights the negative relationship, showing that LEAR works on the single molecule level. The data was only from the $X$ dimension of the dataset.

**Fig I**: **The effectiveness of the methodology given different parameters:** A) Top: the mean distances for the chromatin tracing data of Bintu et al. [10] for the condition labeled; all analysis within this figure uses this dataset. Bottom: The relative error for each loci after the methodology for the amount of localization error introduced. B) The relative error vs the number of loci included in the application of the methodology. C) The relative error as a function of the detection efficiency. D) The relative error as a function of the localization error guess; the true localization error equals 200 nm. E) The relative error vs the number of traces included.

**Fig J**: **Validating the methodology with polymer simulations with structure:** A) The ground truth contact frequency map of loci 700 to loci 900 of the 1 kb polymer simulation data [43]. B) As in the main text the observed 2-way contact frequencies vs. the true underlying contact frequencies for 25 nm or 50 nm localization error both with and without LEAR—again blue line is for reference. C) The same as in B but for all possible 3-way contact frequencies. D) The relative localization error for the individual locus after LEAR for the 25 nm and 50 nm errors. E) The relative localization error for each individual trace.

**Fig K**: **The linkage between Sox2 transcription and the organization of chromatin:** A) An illustration of the *Sox2* region with the two types of FISH signal that Bing et al. used. In this experimental system, a fluorescent protein (FP) sequence was placed after the *Sox2* gene and different probes targeted the *Sox2* RNA or the FP RNA. If both RNAs were detected at a single allele, we refer to that as a 'later burst detection.' If only the *Sox2* RNA was detected at an allele, we call that situation an initial burst, for the RNAP would not have traveled as far after initiation. The 'any transcription' category is for an allele where either a later burst or an initial burst was detected. B) The contact frequency maps for the no boundary condition using the raw data, comparing traces with any kind of transcription (left) to the no transcription frequencies (middle). The difference between the two are shown on the far right. C) The same as *B* but using the raw data boundary condition data. D) The same as *B* but using the methodology. E) The same as *C* but using the methodology.

**Fig L**: **Cooperativity in 4-way contacts is not dependent upon loop extrusion:** A) The 4-way contacts compared to the expected 4-way contact frequency if the 2-way contact frequencies contributed independently, with and without auxin with the application of LEAR. B) The same as *A* but without the methodology.
(PDF)

## Acknowledgments

This work would not have been possible without the computational resources of the NIH HPC Biowulf cluster (http://hpc.nih.gov). We are also grateful for the many discussions with Dr. Nadezda Fursova, Dr. Mohamadreza Fazel, and Dr. Alexander Englert. And thank you to James M. Jusuf for providing raw traces of polymer simulation data. The contributions of the NIH author(s) are considered works of the United States Government. The findings and conclusions presented in this paper are those of the author(s) and do not necessarily reflect the views of the NIH or the U.S. Department of Health and Human Services.

## Author contributions

**Conceptualization:** Christopher H. Bohrer.

**Data curation:** Christopher H. Bohrer.

**Formal analysis:** Christopher H. Bohrer.

**Funding acquisition:** Daniel R. Larson.

**Investigation:** Christopher H. Bohrer.

**Methodology:** Christopher H. Bohrer.

**Project administration:** Daniel R. Larson.

**Resources:** Daniel R. Larson.

**Software:** Christopher H. Bohrer.

**Supervision:** Daniel R. Larson.

**Validation:** Christopher H. Bohrer.

**Visualization:** Christopher H. Bohrer, Daniel R. Larson.

**Writing – original draft:** Christopher H. Bohrer.

**Writing – review & editing:** Christopher H. Bohrer, Daniel R. Larson.

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
