## [Decision Letter · Decision Letter 0]

13 Jul 2025

 PCOMPBIOL-D-25-00741

A methodology to reduce the localization error in multi-loci microscopy provides new insights into enhancer biology

PLOS Computational Biology

Dear Dr. Larson,

Thank you for submitting your manuscript to PLOS Computational Biology. After careful consideration, we feel that it has merit but does not fully meet PLOS Computational Biology's publication criteria as it currently stands.

While all of the reviewers acknowledge the importance of the methodology, several concerns are raised. In particular, reviewer #3 has serious technical concerns about the method itself, which is the primary focus of this work. These will have to be carefully addressed if the work is to be reconsidered. 

Therefore, we invite you to submit a revised version of the manuscript that addresses the points raised during the review process.

Please submit your revised manuscript within 60 days Sep 12 2025 11:59PM. If you will need more time than this to complete your revisions, please reply to this message or contact the journal office at ploscompbiol@plos.org. Please include the following items when submitting your revised manuscript:

We look forward to receiving your revised manuscript.

Kind regards,

Alexey Onufriev

Academic Editor

PLOS Computational Biology

Shihua Zhang

Section Editor

PLOS Computational Biology

**Journal Requirements:**

2) Your manuscript is missing the following sections: Design and Implementation, and Availability and Future Directions. Please ensure that your article adheres to the standard Software article layout and order of Abstract, Introduction, Design and Implementation, Results, and Availability and Future Directions. For details on what each section should contain, see our Software article guidelines:

https://journals.plos.org/ploscompbiol/s/submission-guidelines#loc-software-submissions

3) If any authors received a salary from any of your funders, please state which authors and which funders.

6) Your current Financial Disclosure states, "National Institutes of Health (1ZIABC011383-11)."

However, your funding information on the submission form indicates receiving no funds. Please indicate by return email the full and correct funding information for your study and confirm the order in which funding contributions should appear.

Note : Please ensure that the funders and grant numbers match between the Financial Disclosure field and the Funding Information tab in your submission form. Note that the funders must be provided in the same order in both places as well. 

**Reviewers' comments:**

Reviewer's Responses to Questions

Reviewer #1: The present paper from Bohrer and Larson describes LEAR, a new computational method to reduce the localization error of chromosome tracing experiments, thereby extracting more information.

The is currently great interest in understanding the 3D structure of the genome, its regulation and function. And chromosome tracing is an increasingly common approach to this. Therefore, improving the analysis of chromosome tracing experiments is an important problem, and I believe the LEAR method will be of wide interest in the field.

Beyond the application and description of LEAR, the manuscript also does a great job of educationally discussing the issue of localization error and the consequences (which is poorly understood in the field and the origin of many confusions). The application of LEAR to published datasets is also very helpful, both for the biological insights as well as for illustrating its value.

The LEAR approach appears to me to be sound. I also went to https://github.com/CHB-Bohrer/LEAR/tree/main and although the GitHub repo appears barebones on first encounter, there is a good user guide in the “User_Guide_and_Example_Data.zip” and the Quick User guide seems helpful (the first author even suggests that user text them on the mobile). I think the impact of LEAR will depend on the ease of its use, so making sure that it is very easy to use will be important.

In short, a solid well-written manuscript that should be a wide interest and which seems like a great fit to the journal.

My suggestions are all minor – they do not affect the validity of the approach – but I do think they will be important to address to improve the presentation and balance of the manuscript. Otherwise, I enthusiastically recommend that a suitably revised manuscript be published.

SUGGESTIONS:

1. A discussion of RASER-FISH is missing. Figure 1 in version 1 of the Beckwith 2021 preprint https://www.biorxiv.org/content/10.1101/2021.04.12.439407v1?versioned=true shows very clearly that the chromosome tracing protocol used in all the studies cited in this paper and in all the studies whose data is used here, results in severe perturbations. It is also not surprising. If you boil cells in 50% formamide, it would be surprising if chromatin did not move at least a few tens of nanometers. But it is important to mention this explicitly. I would like to ask the authors to include a brief discussion about de-naturing vs. non-denaturing FISH and the perturbations that results from the standard chromosome tracing procedures. Of course, this is not the authors fault – they are just provided an improved algorithm. But essential to discuss.

2. Building on that, I would like to ask the authors to include a paragraph on chromosome tracing experiment limitations. First one is cell fixation. Second one is denaturing the DNA, by boiling the cells. Beckwith-Ellenberg has clearly shown this. In an ideal work, the authors would also analyze a non-denatured dataset (e.g. from Ellenberg lab) but this likely be beyond the scope of this paper. Third one is sister chromatids. Since most chromosome tracing experiments are done on cycling cells, on average, about half of all chromosomes will be replicated and appear not as a single chromosome but as a pair of sister chromatids. What is then imaged in the chromosome tracing experiment is the average of the 2 sisters, i.e. the average position of the same DNA region in the 2 sisters. Because replicated chromosomes give 2x the signal (after all, they are replicated) any procedure that removes bad data will further enrich for replicated sisters. Accordingly, chromosome tracing experiments are not really tracing single chromosomes. They are reporting average positions between 2 sisters. This is a major source of error, and more or less all chromosome tracing papers sweep this issue under the rug to my knowledge. There are other issues too. I know this is somewhat unrelated to LEAR and LEAR is great, but it will be important to discuss explicitly in a paragraph in the paper for the audience to appreciate other error sources. It is also important to the key question of the correlation between E-P and txn – the authors nicely show that lower error allows making the connection. If these other sources got fixed, the correlation would likely be even stronger.

3. Line 24 cites nice chromosome tracing papers from Nollmann, Boettiger, Zhuang labs, who have done important early work. But there are many many other labs. Chao-Ting Wu did most of the early pioneering work that formed the foundation for the Zhuang work, but many others come to mind like Jan Ellenberg, S Wang etc. Feels slightly unbalanced.

4. This is subjective and minor, but I found the ‘dynamics’ terminology used to be a little odd at times. The first sentence of the abstract says “spatial dynamics”. The fourth sentence “displacements”. These are terms use for tracking experiments like live-cell single particle tracking. But the point of chromosome tracing is that the cells are dead. And static. I suggest to use static terminology for static experiments, and not terms like dynamics and displacement.

5. Small thing but x-axis of Fig S7E should go from 0-100 if it is in %, or did I miss something?

6. It would be helpful to state all the assumptions of the method in one place in the main text. Line S32 in the SI says error need not be Gaussian, but other places say it is assumed to be Gaussian.

7. In line 157 they test LEAR on the tracing data of Huang REF34 and add Loc Error to the data, treating the data as ground truth. This felt a little odd to me, because the data is subject to noise and all the other uncertainties of experimental data. Would it not be much better to run a polymer simulations and generate synthetic chromosome tracing data where the ground truth is known, and then add localization error to the polymer simulations? There should be polymer simulation data available for download from various publications using polymer simulations.

8. LEAR does seem very clear and impressive. But line 211 says “greatly” and then line 214 says “10-20% improvement”, so greatly seems a little overstated.

9. For the discussion around lines 218-224, the clearest demonstration that there is a quantitative relationship between “contact” and transcription to my knowledge is Fig 4D in Kim-2024 https://www.cell.com/molecular-cell/fulltext/S1097-2765(24)00126-6 which the authors could also discuss here.

10. Very subjective, but I do not love “contact frequency” to be equated with “<200 nm threshold”. Contact to me has mechanistic meanings, the two pieces of DNA are in physical contact. A semi-arbitrary distance threshold is a common method of analysis, but in my subjective view, it introduces confusion to equate a threshold to a mechanism. I would suggest to just call thresholds thresholds, and not imbue them with mechanistic meaning.

11. Line 290: makes sense since loop extrusion is at its core a 1D mechanism.

12. Fig 5A color bar, ticks are missing a label. Is if 0.4% or 40%? I assume the latter.

13. Line 318, “inversely proportional to the distances between loci” – can you state whether this is distance in nm or kb.

14. Line 362-364, yes, but also boiling cells, fixation artifacts, sister replication, etc.

15. MINFLUX is awesome, but is unlikely to take over any time soon due to practical limitations. I almost feel like the authors undersell LEAR here.

Reviewer #2: Bohrer and Larson introduce a post-processing computational approach to minimize the contributions of measurement error. This work offers the first systematic application of measurement error quantification (already systematically included in many datasets of this sort), to improve the quality and interpretation of the data. Current approaches by contrast have largely relied on these error measurements as control data. Leveraging this information, the approach also offers the first (to my knowledge) attempt at systematic error correction based on a minimal set of assumptions. Overall the manuscript is well written and represents an important advance in a rapidly growing field. I would support publication, though I recommend the authors consider the issues I discuss below in a revised manuscript.

The iPython notebook runs on the included demo data without problem, and produces the expected results. The software and demo data were easy to download and the demo easy to navigate. These are critical features for a software paper and a disappointing number of articles reach this standard on the first go so I commend the authors there.

My primary recommendation would be to test the method on some fully simulated data, which may help provide a better estimate of the maximum improvements that may be achieved by the method (given the logical arguments the authors present about why existing data may underestimate the approaches performance), as well as better identify potential additional artifacts the method might introduce – see below for details. I include several other suggestions for improving the clarity and potentially the rigor of the analyses below, and I apologize in advance for the lack of brevity in these comments, I lacked the time to write a shorter letter.

Primary recommendations:

I think one fully-simulation based dataset + simulated noisy acquisition + method application would be interesting to test the upper limits of the correction, and to identify potential sources of error introduced by the method. I appreciate the power of adding noise to current data as a way of keeping a truly realistic background model, I would not suggest removing that. But there is still much more uncertainty in understanding which features of the data are true variation and which are arising from the noise when using real data.

For example, even in the original filtering steps, based-off analyses of the replicate data, the simulations will show what sort of errors are introduced and what frequency by hard thresholds (below the maximum stretch length for the underlying polymer) for the max distance to neighbors.

Further, the authors identify a series of quite logical arguments why the true improvement achieved by the method may be greater than those estimated by comparing real data vs real data+ noise or by comparing two noisy replicates to each other. This ‘true improvement’ (or at least a better understanding of the maximum potential improvement), could be better understood and quantified with a fully known, simulated ground truth.

There are numerous tools available for doing polymer simulations these days, I might recommend some some of the loop extrusion simulations from the Mirny lab or other open2c/polychrom contributors.

Figure 2 I think could use some improvement, as I missed the message of this figure on the first read and was going to suggest moving the entire thing to supplement. I think instead some reorganization and modification to the plots would make the message clearer.

2.1 In (iii), is “relative error (STD)” = error with LEAR-corrected data / error of orig data? If so I would label it as something such, maybe make it on log2 axes and maybe leave the above zero part of the graph blank so that it is more apparent at a glance that the method reduces the error, particularly improving Z. Currently I have to stare quite long to figure out what’s going on in this figure, and closely examine the tiny numerical values on the y-axis to tell that any improvement has occurred.

2.2 It would also be instructive I think to see the distance maps (or contact maps) before and after the method. I’d be curious if the correlation of the corresponding contact-frequency maps with bulk Hi-C changed at all as well in the before vs. after the correction. As long as the Hi-C data isn’t referenced in whatever correction approach is being used (as is the case here), it is a nice orthogonal way to cross-validate the data and validate the correction.

2.3 The panels (ii) are a bit of technical detail specific to the datasets (what their actual errors were before and after, I also can’t tell from the text if this is before or after applying the method). Maybe these could go into a supplement to make room for some more maps.

2.4 On first read I got more out of Fig S2, which shows how different data properties (like number of loci, detection efficiency, sigma-guess) effect the ‘relative error’. After deeper study I think Fig 2 panel (iii) have important data relevant to the main claims of the paper, but (i) and (ii) are less informative that the graphs in Fig S2. The authors might consider moving some of these data up to the main Figure.

Minor recommendations:

How much of the effect of LEAR shown in Fig 3 and 4 is explained by simply reducing the 3D distances through the ‘correction’ process. The increased pairwise and 3-way contact frequency after error addition in Fig 3 H and I with a fixed contact threshold is also expected if one isotropically expands ‘correct’ data. Similarly with the increase in contact frequency in the Huang data with and without the CTCF insertion. Do the average 3D distances get smaller? If so by how much? I appreciate a compaction effect wouldn’t explain all of the improvements (such as those shown in fig 2, or increasing the measured strength of insulation), but some supplement on the effect on total average distance (e.g. radius of gyration) and the contribution of any systematic change in this average to the changes in contact threshold could be addressed.

Multi-way contacts +/- cohesin

With LEAR correction, can you see an effect on cooperative contacts at CTCF sites (similar to that reported by Hafner et al 2023 (Hafner et al. 2023), and multi-contact 4C (Allahyar et al. 2018))

The subsection title is a bit problematic - it would read better as ‘not the only mechanism’ rather than ‘not the mechanism’.

In addition to being overstated, the ‘not the mechanism’ comment is very much in line with the picture presented by the Bintu, Mateo, 2018 data, which shows little change in the ‘cooperative’ 3-way contacts upon cohesin degradation (though I appreciate the additional quantification approach used here). The 4-way interaction seen after the correction, for example, is more novel.

The discussion of this result (in the main text, before the discussion section) seems a little under-developed. It is unclear if the authors are concluding that the effect on 4-way contacts is purely an effect of increased 2-way contacts, or if they are suggesting that an inherent proximity-dependent cooperativity among chromatin contacts is amplified by the contact.

Fig S6 – The layout could be improved.

The regions in A and C could be shown as zoom ins on E so that the comparison would be easier for the reader. Especially lacking axis labels in E it is hard for those not familiar with the size and coordinates to match these up.

Data filtering

While a very elegant argument is outlined on why some data points are erroneous (those that were repeat labeled and gave two very different answers, or those that were repeat labeled and gave distances far from the nearest neighbors both times, the authors filtering solution to address this is rather buried in the supplement, and appears to be a bit simplistic (which is certainly meritorious in easy-to-apply, but maybe could be made a little more precise). On one hand I thought this was an important insight that merited a clearer explanation and discussion in the main text (just a couple more sentences would go a long way). On the other hand I think a few more controls in the supplement and a more algorithmic way of settling on the final cut-off (or an alternative to what I understood as a simple hard threshold) would also be nice.

4.2 Additionally, it was unclear how exactly the thresholds were applied. There appear to be hard cut-offs: 500 nm (or 400 nm) in the nearest neighbor jump – what should one do about points with missing data? E.g. 2 points missing between, allow 1000 nm jumps?

4.3 Can one do better than a hard cutoff? Based on polymer theory, the distance between two points have heavy tailed distributions with rather long tails. The precise shape is affected by details of the theory, e.g. crumpled/unknotted, vs. equilibrium globule vs. dilute/Rouse polymer – but all have finite support (e.g. a hard maximum cut-off at the truly stretched polymer). Notably this polymer stretched distance: 50 kb at 10 nm/200 bp = 50 nm per kb = 5000 um truly stretched, so achieving 1000 nm end-end is not too much of a stretch, though empirical data show this is very unlikely. I think the physically realistic hard-cut off at some estimate of the maximum stretch of the polymer is too lenient, but a truly hard-cutoff is rather non-physical (i.e. it would exclude some error-free data), though likely not exclude enough data to make an practical difference. Can one do some probabilistic rejection of data based in its observed vs. expected frequency from a polymer model?

4.4 I don’t think it’s necessary for the authors to be apologetic about the need for filtering, though I very much appreciated the analysis of the justification for filtering. I think this could fairly be identified as an equally important step as “denoising” that is achieved by the core LEAR algorithm, and a subject matter in need of more rigorous intellectual engagement in the community, which has previously been handled on a very ad hoc basis. It could be instructive also to compare prior ad hoc decisions to the more data-driven, systematic approach used here.

For example, prior work using the Bintu, Mateo, 2018 (Bintu et al. 2018) data vary dramatically in which traces they actually use: (Goundaroulis et al. 2019) toss all but 243 of the over 4,500 traces when using the data to say something about knots in the path. By contrast (Conte et al. 2020) toss way fewer traces from these data using a different set of logical sounding but equally ad hoc criteria: “Here and in the following analyses, we filtered out the experimental single-cell distance matrices having NaN values for more than 80% of the entries and, in order to remove outliers, the matrices having a Pearson correlation <0.01 with the others are also removed in both models and experiment”

These data admittedly contain some dubious long range jumps. However, these jumps are not so long as to be categorically ruled out on simple physical properties, such as the maximum stretch distance of DNA however, nor are they filtered by brightness. Comparison to Hi-C provides an orthogonal measure that indicates some barcode positions appear to be noise uncorrelated with the the orthogonal approach, whereas most of the data correlate strongly (and exhibit other properties expected of a polymer, such as the preference to be nearer in 3D space to sequences known to be closer in sequence space along the polymer). It’s les clear that there’s any physical justification for rejecting outliers based on Pearson correlation or % missing data (missing data is likely a stochastic event driven by stochastic properties of the a labeling approach that relies on chance to reduce a duplex strand of DNA to a single copy by randomly chopping and melting out bits, hoping that 1 is removed not both). Such a process is not expected to give uniform coverage in each trace, nor are data from higher-miss-fraction traces necessarily an ‘worse’ a measure of reality.

In contrast to unjustified ad hoc approaches, these authors provide a more rationale, data driven explanation of which data to drop. This important contribution could be expanded, and maybe some more general guidelines could be derived from this analysis for future work. Such guidelines could also be used in iterative fitting algorithms – in practice multiple potential ‘spots’ can be detected in a single image and the software could use some guide on which ones to choose (e.g. brightest is not always best).

Minor Text issues to clarify

“the number of loci that were repeatably imaged, with the 50kb resolution data having the most repeatably imaged loci ≈ 20”

I think some further unpacking of what is meant by “the number of loci that were repeatably imaged” is necessary. I suspect many readers will mistakenly interpret this as the number of loci that could be reliably detected across cells. I believe what the authors mean is the number of loci that were imaged multiple times within each cell for the purpose of quantifying measurement error.

[intro] “. While there is an inherent uncertainty within all fluorescence imaging techniques interrogating spatial features less than the wavelength of visible light,”

I’d reword this. There is extensive confusion in the scientific community about the relation of the diffraction limit and multiplexed imaging - with many scientists incorrectly associating the Abbe diffraction limit with ability to resolve two fluorescently labeled spots, labeled with chromatically distinct fluorophores or in sequential rounds with same fluorophore. As the current authors I am sure appreciate, if two elements of DNA are labeled with fluorophores of the same wavelength and imaged at the same time, there is indeed a diffraction limited resolution problem. If the fluors are different colors (or in different rounds) the *wavelength of visible light* has a minimal effect on resolution that is almost always dominated by 1 of dozens of other sources of uncertainty.

It would be equally correct to write that STORM and PALM are limited by the wavelength of light. While technically true, this would be unnecessarily confusing to the reader, as the wavelength limit is too intricately connected with the idea of the diffraction limit, and as other sources -(quantum efficiency, photon shot noise, duty cycle) typically dominate the contributions of the wavelength of light in chromatin tracing data (as they also tend dominate most STORM and PALM data and related SMLM techniques).

Fo what it’s worth: I’m pretty convinced the primary limitation to resolution in chromatin tracing comes from the size of the DNA labeled in most data sets. This region is 2 kb in the highest resolution traces, and more often 10 kb or 50 kb in much of the high quality public data. This distance is not a point source, and at ~ 10 nm/nucleosome, the finite size occupied by a flexible chromatin polymer of 10-250 nucleosomes can easily be on the scale of the distances measured in the microscopy (50-1000 nm is a typical range, and 2 kb can be 100 nm and 50 kb can be 1000 nm though it averages closer to 150 nm). The labeling procedure is also not uniform, since it relies on stochastic elimination of DNA to produce single-stranded DNA for binding (transient DNA melting is insufficient for labeling, as the complementary strand is longer than the probe, it can always undergo a toe-hold mediated strand-displacement reaction to evict a fluorescent oligo, unless it is actually cleaved and released from the cell or degraded). The data analyzed here was all created using acid and heat denaturation, which I believe stochastically nicks the DNA backbone allows short strands to drift away. Consequently a labeled 10 kb region will consist of variably sized lengths of actually exposed target (including some cells with no exposed target, hence the drop-out rate, even when imaged in thin sections with cameras that have no difficulty detecting single cy5 fluors, the detection efficiency for short regions is never 100%, and it increases with longer regions as the probability of retaining the target DNA so there’s something for the oligo to label in the first place increases). This stochastic variation in the size and position of what is labeled, cell-to-cell, further contributes to an uncertainty in the interpretation of the population level distribution of the 3D distance between elements. – The elements themselves actually have some positional variation due to the stochastic labeling and these labels are distributed over a polymer of finite size, not a point source.

For some datasets, drift correction can be a significant limitation to resolution as well - many studies use beads to track the position of the coverglass to correct stage drift, but this contains no information about minor changes in the position of individual cells (or individual chromosomes), which may change subtly over days of repeated washes and imaging. Most approaches use z-scans, and the return accuracy and alignment of z-position does not exceed 50 nm precision on all systems unless some explicit care is taken. For computational efficiency, rounding some analyses rely on image based correlation analysis to correct drift, in which images are aligned on finite sub-pixel shifts (e.g. 25 nm in Mateo 2019), setting another hard-limit to resolution accuracy.

4. [abstract]: “we then applied our approach to existing chromatin tracing data that probed the relation between chromatin organization and Sox2 regulation, where previous work found no correlation between enhancer-promoter proximity and transcription bursts in individual cells.” See ALSO:

[intro] “no direct relation to nascent activity was found in individual cells”

I think this is an overstatement, I believe there is a correlation in these data without correction (from our own plotting of these data without correction, as well as based on statements in the original article). Specifically:

“the median enhancer–promoter distances at the Sox2 gene with coincident nascent transcripts were slightly but significantly shorter than those on the resting loci (Extended Data Fig. 10c,d).” - see ED Fig 10d of ref 32.

“These results, taken together, suggest that enhancer proximity is positively correlated to transcriptional activity at the target gene” (ref 32)

While I believe the original data showing the (weak yet statistically significant) correlation could have been shown and explained better, I think it is inappropriate for the current work to say no correlation was found or to conclude no direct relation to nascent activity in individual cells was found. I don’t have a problem with the authors saying the relationship was previously “obscured by localization error”, (line 67) to me this doesn’t imply it was totally invisible and concluded not to exist. I’d recommend a phrasing of this sort also be adopted for the abstract.

5. Typo: Line 340, “Interestingly, due to the cooperatively, higher-order multi-way contacts could be essentially…” should say “cooperativity”

Reviewer #3: This article by Bohrer and Larson introduces a new methodology, LEAR, to reduce localization error in chromatin tracing experiments (i.e. collections of 3D conformations of a portion of a chromosome, obtained by sequential DNA FISH microscopy). This approach relies on an estimation of the localization error and on the realization that, for each imaged DNA locus, its pairwise distance distributions with all the other DNA loci in the collection of conformations bear information that can be harnessed to reduce localization error effects. Using publicly available datasets, the authors evaluate how LEAR performs at restoring artificially-deteriorated experimental data and then revisit questions about enhancer-promoter contacts at the Sox2 locus and multi-way chromatin contacts with and without the cohesin complex.

I find the motivation strong and the approach original, but I have several concerns — especially about the interpretability of the “corrected” data, the limited value of the validation strategy, and the clarity of the manuscript. Below I outline the most important points. The application to experimental data illustrates the method but yields modest biological insight.

*** Major comments ***

1. A central concern is that the method cannot meaningfully correct individual traces. For each locus $\alpha$, the method estimates the N correction values based on only L−1 constraints (i.e. vector $C^\alpha$, deduced from $Var(T^\alpha-O^l)$ for all the loci $l$ other than $\alpha$). Since N >> L in typical datasets, the problem is severely under-constrained. The manuscript gives the impression that individual corrected traces $A^\alpha$ are meaningful, which they are not. Instead, the only meaningful outputs are ensemble-level statistics, whether they are pairwise (contact or distance matrices) or higher-order (e.g. 3-way) statistics. At the level of single traces, any additional structure created by the correction procedure is artificial: the true conformation cannot be reconstructed from the available data, and no trace-level information has been gained. It is essential that this be clearly stated to avoid misinterpretation by future users.

2. The part on “simulated data” is unconvincing. The authors do not generate synthetic data with known structures. Instead, they take experimental data, apply LEAR once and treat it as “ground truth,” then add noise and check whether LEAR can subtract it back. This is a peculiar way to test a method. It lacks realism and fails to assess whether LEAR can recover biologically meaningful features. A more convincing validation would involve polymer simulations (e.g. with loop extrusion) or even simple random conformations that contain by construction features such as looped structures or enriched 3-way contacts.

3. The article is difficult to navigate due to its structure and referencing. In the main text, figures S4–S8 are mentioned before S1–S2, and S3 is never cited. The “Supporting Material” is often referenced without clear indication of which section is meant, and within it, forward and backward references are frequent and vague (“see below,” “discussed more later”), making it hard to follow. There is no clear way to navigate all this material. I list this as a Major Comment –for a reader trying to understand the method, this maze-like structure is a major obstacle.

*** Minor comments ***

4. A key limitation is that LEAR can only be applied if $Var(\epsilon^\alpha)$ is known. This requires re-imaged loci (or strong assumptions about what $Var(\epsilon^\alpha)$ would be in an experiment where loci were not re-imaged). The manuscript should make this clear (and from the start) to avoid giving a false sense of general applicability.

5. Regarding the last analysis of the Bintu et al. datasets, a simple hypothesis to explain the observed correlation between contact frequencies of distinct locus pairs (what the authors call “cooperativity”) is global variability over the observed region. For example, cells might be in different states (e.g. cell cycle phases), or the entire region might experience global effects such as changes in nuclear positioning or variations in local nuclear context that affect overall compaction. This or anything that affects the observed region globally would naturally lead to correlated contacts between all locus pairs, which would manifest as elevated 3-way contact probabilities that deviate from expectations assuming independent on 2-way contacts. This simple scenario would indeed be independent of loop extrusion and should be discussed as a plausible null model.

6. In the paragraph around line 94 of the main text, it would help to clarify that $Var(T^\alpha - O^\beta)$ is discussed as a distribution because of the finite number of traces, i.e., it refers to sample variance. Earlier, the term “variance” referred to population variance. Using this standard terminology (sample variance and population variance), here and in the Supplements, would be preferable and would improve clarity.

7. And using known results about sample variance distributions: To estimate $\sigma_{\alpha,\beta}$, instead of using Algorithm 2, couldn't one use the fact that sample variance follows a chi-squared distribution? That would give $\sigma_{\alpha,\beta} = \mu_{\alpha,\beta} \sqrt{2/(N-1)}$. At the very least, this should be discussed.

8. The phrase “we can determine the ground truth” (abstract) seems too strong. Inverse problems by construction only allow us to infer or approximate a ground truth, not determine it.

9. In Equation (2), the “probability” is actually a probability density.

10. Figure 2 would benefit from explicitly stating what “relative error” refers to.

11. In Algorithm 2, it would improve readability if you used the same notation as in the main text: using $C^{sim}$ rather than $\epsilon^{sim}$.

12. For comments, Algorithm 1 uses “%” and Algorithm 2 uses “#”.

13. In the section that start on page 4 of the Supporting Material, why "$Contant$"? Did you mean "$Constant$"?

I used a generative AI tool to assist in improving the clarity and structure of this review. All scientific assessments and opinions are my own.

**Have the authors made all data and (if applicable) computational code underlying the findings in their manuscript fully available?**

Reviewer #1: Yes

Reviewer #2: Yes

Reviewer #3: Yes

PLOS authors have the option to publish the peer review history of their article (what does this mean?). If published, this will include your full peer review and any attached files.

Reviewer #1: No

Reviewer #2: No

Reviewer #3: No

**Figure resubmission:**
---

## [Decision Letter · Decision Letter 1]

28 Nov 2025

Dear Dr. Larson,

We are pleased to inform you that your manuscript 'A methodology to reduce the localization error in multi-loci microscopy provides new insights into enhancer biology' has been provisionally accepted for publication in PLOS Computational Biology.

Before your manuscript can be formally accepted you will need to complete some formatting changes, which you will receive in a follow up email. Also, please address the additional (minor)  suggestions made by reviewer #3.  A member of our team will be in touch with a set of requests.

Best regards,

Alexey Onufriev

Academic Editor

PLOS Computational Biology

Shihua Zhang

Section Editor

PLOS Computational Biology

Reviewer's Responses to Questions

**Comments to the Authors:**

Reviewer #1: The authors have, for the most part, addressed my comments. It's nice to see the inclusion of the ground truth polymer simulation data, and it seems like this was requested by all the reviewers. My only remaining question is why the contact map in Fig 4A looks "structure-free" (as in no triangle, and no dots, unlike in Jusuf et al.). If I am confused, other readers might also be, so it may be worth clarifying this.

Otherwise, i think this is an excellent paper that is likely to be a wide interest and the authors generally did a great revision. I appreciate the attention to rigor and making the method easy to use. I anticipate and hope that this method will be widely used by the chromosome tracing community. I recommend publication without any further delay.

Reviewer #2: The authors have addressed my concerns. The additional analysis of the simulated data I believe strengthens the paper and nicely illustrates the improvement.

I think the additional figure could benefit from an added panel showing an example image of an original, ground-truth 3D trace, the noise-added version of the trace, and the LEAR corrected version of the trace. There appears to be some extra whitespace in this figure compared to the others, and I think such an image would be both illustrative of the improvement and improve the readability of figure.

Reviewer #3: In this revised version the manuscript, the authors have appropriately addressed most of my comment. As I detail below, I am now convinced that my major concern (point 1) was incorrect. I nonetheless suggest an additional plot to make it more clear to the readers. I also suggest an improvement on what is shown using polymer simulation data. But these are only minor points. I consider the manuscript appropriate for publication.

Regarding my points 1 and 2:

1. Indeed, I am now convinced that the authors are right and that LEAR does apply corrections that are meaningful at the individual trace level, not just the ensemble level. Something that contributed to convincing me and that would be worth including (and emphasizing) in the article is the following:

After re-running the code, I made the scatter plot of C as a function of epsilon (i.e. the correction that LEAR subtracts from the observations $O$ in order to approximate ground truth $T$, as a function of the error that was added to ground truth $T$ to obtain the 'observations' $O$). If LEAR is able to correct localization errors at the single trace level, these two quantities should be negatively correlated with a slope -1. We do see that they are indeed negatively correlated, but the cloud of points has an interesting shape, which would be interesting to comment.

Here is the minimal code to reproduce this (e.g. in the Simple_Analysis.ipynb notebook provided on GitHub, just run the first two cells, then run the following):

T,O,A = [np.array([np.load(path+f"{i+1}.npy") for i in range(1390)]) for path in ['Ground_Truth_Trajectories_True_X_129/', 'Trajectories_Error_X_129_N100nm/', 'Corrected_Trajectories_True_X_100/']]

epsilon, C = O-T, A-O

plt.figure()

plt.scatter(epsilon, C, alpha=.1, s=5)

plt.plot([-150,150],[150,-150], color='k')

plt.xlabel("$\\epsilon$, error added to ground truth to obtain observations (nm)")

plt.ylabel("$C$, correction subtracted from observations\nto approximate ground truth (nm)")

plt.show()

2. It is nice that the authors included polymer simulations as a ground truth data set to test LEAR. They reused simulations from Jusuf et al., but why did they deliberately chose a completely featureless region (i.e. the first 100 loci, where no CTCF site are included)? To be convincing, they should have shown a region rich in CTCF sites and loops and show how applying LEAR recovers features that are lost in a deteriorated version of the data.

All of my other points have been properly addressed.

**Have the authors made all data and (if applicable) computational code underlying the findings in their manuscript fully available?**

Reviewer #1: Yes

Reviewer #2: Yes

Reviewer #3: Yes

PLOS authors have the option to publish the peer review history of their article (what does this mean?). If published, this will include your full peer review and any attached files.

Reviewer #1: No

Reviewer #2: No

Reviewer #3: No

---

## [Editor Report · Acceptance letter]

PCOMPBIOL-D-25-00741R1

A methodology to reduce the localization error in multi-loci microscopy provides new insights into enhancer biology

Dear Dr Larson,

I am pleased to inform you that your manuscript has been formally accepted for publication in PLOS Computational Biology. Your manuscript is now with our production department and you will be notified of the publication date in due course.

With kind regards,

Anita Estes
